# An Italian Survey on Dietary Habits and Changes during the COVID-19 Lockdown

**DOI:** 10.3390/nu13041197

**Published:** 2021-04-05

**Authors:** Luana Izzo, Antonio Santonastaso, Gaetano Cotticelli, Alessandro Federico, Severina Pacifico, Luigi Castaldo, Annamaria Colao, Alberto Ritieni

**Affiliations:** 1Department of Pharmacy, University of Naples “Federico II”, Via Domenico Montesano 49, 80131 Naples, Italy; alberto.ritieni@unina.it; 2Department of Precision Medicine, Hepatogastroenterology Division, University of Campania “Luigi Vanvitelli”, Via Pansini 5, 80131 Naples, Italy; antonio.santonastaso1@studenti.unicampania.it (A.S.); gaetano.cotticelli@unicampania.it (G.C.); alessandro.federico@unicampania.it (A.F.); 3Department of Environmental, Biological and Pharmaceutical Sciences and Technologies, University of Campania “Luigi Vanvitelli”, Via Vivaldi 43, 81100 Caserta, Italy; severina.pacifico@unicampania.it; 4Department of Clinical Medicine and Surgery, Unit of Endocrinology, University Federico II, Via Sergio Pansini 5, 80131 Naples, Italy; annamaria.colao@unina.it; 5UNESCO Chair on Health Education and Sustainable Development, “Federico II” University, 80131 Naples, Italy

**Keywords:** COVID-19, coronavirus, dietary habits and changes, MEDAS score, Mediterranean diet adherence

## Abstract

The World Health Organization has declared the coronavirus outbreak a Public Health Emergency of International Concern; the outbreak has led to lockdowns in several parts of the world, and sudden changes in people’s lifestyles. This study explores the impact of the first coronavirus disease 2019 (COVID-19) pandemic period on dietary habits, lifestyle changes, and adherence to the Mediterranean diet among the Italian population, through an online questionnaire, conducted from April to May 2020, involving 1519 participants. The 14-point Mediterranean Diet Adherence Screener (MEDAS) highlighted a medium Mediterranean diet adherence in 73.5% of responders, which principally included the younger population, aged 18–30 years (*p* < 0.05). In regards to changes in eating habits, 33.5% of responders declared an influence of the pandemic period on nutritional practice. A decrease in alcohol consumption was reported by 81% of responders, while an increase in frozen food consumption was reported by 81.3% of responders. In addition, 58.8% reported positive weight modification (40.8%, +1–3 kg); physical activity reduction was reported for 70.5% of responders. Our study contributes toward amplifying the investigation on the dietary habits and changes of the Italian population during the COVID-19 lockdown, although the pandemic is ongoing. Similar studies should be performed around the world to understand how the emergency has impacted people’s habits.

## 1. Introduction

The World Health Organization has declared the coronavirus disease a Public Health Emergency of International Concern (PHEIC). The outbreak has led to lockdowns in several parts of the world, sudden changes in people’s lifestyles, and economic and social consequences [1]. Severe acute respiratory syndrome coronavirus 2 (SARS-Cov-2) is the virus that causes COVID-19; it emerged for the first time in December 2019 in Wuhan City, Hubei, central China [2].

The virus that causes the infection belongs to the coronavirus family; it is highly contagious, can spread among people by indirect, direct, or close contact, through nose and mouth secretions, and is released when an infected person speaks, coughs, or sneezes [3].

COVID-19 has spread all over the globe. Europe reported more than 28% of global COVID-19 infections, of which, a high number of cases were reported in Italy, especially in Lombardy, Veneto, Piedmont, Emilia Romagna, Campania, and Lazio regions. The percentage of other Italian regions (as of January 2021) did not exceed 6% of cases [4].

A graphic representation of Italian COVID-19 infections is shown in Figure 1.

Considering the absence of valid pharmaceutical prophylaxis and treatments (in particular, vaccines, which have been available from December 2020), several countries have opted for strict lockdown measures. The mass isolation approach appeared for the first time in the 14th to 15th century, during which bubonic plague killed close to one-third of the western European population. The term “quarantine” is derived from the Venetian dialect and means “forty days”, to define the strategies of isolating persons with the disease from unaffected persons. Just as quarantine measures were used to fight the Black Death, similar stringent measures have also been applied to help fight the spread of COVID-19 [5].

On 30 January, 2020, the World Health Organization (WHO) declared the novel coronavirus outbreak a global health emergency [6] due to the risks the virus posed to countries. On 11 March, 2020, the WHO declared the COVID-19 outbreak a pandemic. For precautionary purposes, the Italian government set stringent containment measures [7]. In Italy, the Decree “#Io resto a casa” instructed people to stay at home and only go out if strictly necessary. This involved a drastic reduction of any form of socialization [8]. During the quarantine period, the population was required to stay at home and only go outside in close proximity for essential needs, limiting social gatherings and physical activity. For some job categories, excluding essential sectors (e.g., food, pharmaceuticals), teleworking was encouraged in order to limit spread of the virus [9]. 

At the beginning of May 2020, restriction measures were partially lifted, but this unprecedented situation has disrupted peoples’ daily routines and nutritional behaviors [10]. For example, the restrictive conditions made following a healthy and balanced diet difficult. To avoid continuous access to supermarkets, people stockpiles food, which limited consumption of fresh foods, especially fruits, vegetables, and fish. An increase in the purchase of processed food, snacks, junk foods, and ready-to-eat foods was registered [11]. Concurrently, a significant increase in online shopping was recorded (+57% in the penultimate week of February, +81% in the last of February, and +97% in the second week of March). A reduction of purchases in markets that were too crowded and less safe than smaller stores was also observed. The stringent limitations favored spending in nearby stores, increasing purchases +20% compared to the previous months, with evident growth. A progressive reduction in the growth of local markets was also observed—many of which closed due to government limitations [12].

Recent studies have reported that confinement was often associated with a marked change in lifestyle. Some COVID-19-related behavioral changes were observed, including a greater energy intake, as food tends to comfort and support people in stressful conditions (e.g., more boredom, more alcohol intake, and less physical activity). Foods rich in simple carbohydrates have a positive effect on mood and reduce stress due to serotonin production [13,14]. In addition, people preferred homemade sweets and savory products, as takeaway was not available. Eating became a way to spend free time. Considering the long time it will take for the virus to subside, its impact on lifestyle-related behavior is destined to become significant [15].

Eating and lifestyle habits, as well as a balanced diet, play a crucial role in the prevention of some of the most common diseases in industrialized countries, such as diabetes, gastrointestinal disorders, obesity, hypertension, and pathologies related to metabolic syndrome [16]. A comprehensive report from the World Obesity Federation highlighted linear correlations among COVID-19 mortality and the proportion of adults that are overweight. COVID-19 death rates were 10 times higher in countries where more than half of the population was classified as overweight [17]. A healthy diet is important in modulating the processes of inflammation and oxidative stress [18]. Several studies have reported a direct association between adherence to the Mediterranean Diet (MD) and a reduction in overall cancer-related mortality. The Mediterranean diet is the typical dietary pattern consumed around Mediterranean regions, including those in southern Italy [19]. The Mediterranean diet has gained the attention of the food–science community as it reportedly provides protection against chronic disease, and contributes toward favorable health. In recent decades, several epidemiological and experimental studies have looked at the beneficial role of the Mediterranean diet in reducing the incidences of atherosclerotic clinical events in populations with Mediterranean dietary patterns [20,21]. Essentially, the traditional Mediterranean diet is characterized by a high intake of fruits, vegetables, cereals, legumes, nuts, and seeds; with olive oil as the most common source of monounsaturated fatty acids; and low to moderate intake of fish, poultry, and red meat, a low intake of dairy products (principally cheese and yogurt), and light–moderate wine consumption [22,23].

Hence, the aim of the present study is to explore the impact of the COVID-19 pandemic on dietary habits, lifestyle changes, and adherence to the Mediterranean diet among the Italian population, to explore its associated factors. This was done via an online questionnaire during the isolation period (due to COVID-19).

## 2. Materials and Methods

### 2.1. Study Design, Distribution, and Collection of Data

The project “Eating Habits and Lifestyle Changes in COVID-19 lockdown” (Eating Habits Lifestyle Changes (EHLC)–COVID-19) was designed and conducted during the COVID-19 lockdown period from 5 April to 5 May, 2020, by using a web survey. The online platform, SURVIO, was easily accessible through any device with an Internet connection. The link to access the survey, together with a brief explanation of the project, was sent via email, or spread and shared over common social media networks, including Facebook, WhatsApp, and Instagram, in the Italian population. Data were collected online by the system and were then analyzed. At the end of the online questionnaire, each participant provided informed consent. The full version of the questionnaire is available in the Appendix B.

### 2.2. EHLC-COVID19 Questionnaire

The complete questionnaire was divided into three sections: (1)personal data, such as, demographic questions, level of education, and diagnosed pathologies;(2)Mediterranean Diet Adherence Screener (MEDAS) score;(3)dietary habits and lifestyle changes during the COVID-19 pandemic period.

Adherence to a Mediterranean diet was assessed with the MEDAS questionnaire from the PREvención con DIeta MEDiterránea (PREDIMED) study, a primary prevention nutritional intervention trial [24]. MEDAS is a 14-point questionnaire, which includes 12 questions on food consumption frequency and 2 questions on food intake habits related to the Mediterranean Diet (Appendix: Questionnaire) [25]. A value of 0 or 1 was assigned to each question; the value 1 was applied when the adoption of the Mediterranean diet was met, while the value 0 was assigned when the condition was not met. The MEDAS score (sum of the above items) ranged between 0 and 14. Based on the MEDAS score, the studied population was divided into three classes: low (score ≤ 5), moderate (score between 6 and 9), and high (score ≥ 10) adherence to the Mediterranean diet.

### 2.3. Statistical Analysis

Continuous variables were expressed as mean ± standard deviation and categorical variables as number and percentages. Normal distribution was found for all the variables according to the Shapiro–Wilk test. Mann–Whitney U and Kruskal–Wallis tests were performed to compare continuous variables among two or more groups, respectively. All statistical tests were significant for *p*-value < 0.05. Statistical analysis was performed using STATA 12 (StataCorp LP, College Station, TX, USA).

## 3. Results and Discussion

### 3.1. Study Design and Participants

At the end of the web survey, which was spread throughout the Italian regions, including the islands, the obtained data were collected and analyzed. In 2058 participants, 1519 completed the questionnaire and were included in the present study. Participants were divided into four different age groups: 1.6% were in the age group 0–17, 54.1% in the age group 18–30, 41.7% in the age group 31–54, and 2.6% in the age group >55. Of the total participants, 71.6% were females; to overcome the unbalanced gender differences in the studied population, the background characteristics of participants were divided by gender. As shown in Table 1, greater Mediterranean diet adherence was shown in the population group aged 18–30 years (*p* < 0.05). These results confirm and extend previous findings, indicating that this age group shows higher adherence to the Mediterranean diet than the elderly and younger populations [26,27].

On the other hand, the MEDAS score showed significant differences across education levels. Study participants had varying education levels: 23 (1.5%) participants had a secondary school certificate, 630 (41.5%) participants had a high school diploma, and 866 (57.0%) participants had a degree—masters or PhD. The results are summarized in Table 1. Participants in the uppermost education level showed higher adherence to the Mediterranean diet (*p* < 0.05). Our results are consistent with those reported previously, where low education was linked to poor adherence to Mediterranean-like eating patterns [28].

In the first part of the survey, participants were asked to indicate their diagnosed pathologies. Cardiovascular, cancer, diabetes, thyroid disorder, gastrointestinal, and other disease were reported by 32.1%, 24.2%, 7.3%, 6.2%, 4.9%, and 4.5% of participants, respectively. The obtained percentages are in accordance with the Italian trend. In 2014, two-thirds of all deaths in Italy were ascribable to either cardiovascular diseases (40%) or cancer (24%) in women, and one-third of deaths for men. Lung cancer remains the main cause of cancer mortality, followed by colorectal cancer, breast cancer, and pancreatic cancer. Data from the European Health Interview Survey (EHIS) reported that around 5% of Italians were affected by asthma, more than 20% lived with hypertension, and around 6.5% had diabetes. It would seem that individuals with the lowest level of education are twice as likely to live with hypertension, three times as likely to live with diabetes, and four times as likely to live with depression [29].

### 3.2. Adherence to the MD

In the study reported herein, the MEDAS score was utilized to assess the Mediterranean diet adherence during the COVID-19 lockdown in the southern Italian population. The responses to the 14-item MEDAS questionnaire are listed in Table 2.

In general, olive oil represents the main culinary fat consumed (92.9%), and one out of two participants used 4 tablespoons or more. Moreover, a high consumption of legumes, nuts and, vegetables (52.4, 64.8, and 52.5, respectively) was observed in the investigated population; these food products may support the immune system and help to reduce inflammation [30]. As legumes and fish are excellent sources of high-quality protein and low in fats, the inclusion of these food groups should be encouraged to provide more diversified and balanced diets rich in nutrients, especially during pandemics such as COVID-19 [31]. However, the consumption of fruits and vegetables was still below those recommended by dietary guidelines. During the period of confinement, limited access to daily grocery shopping provoked a reduction in the consumption of fresh foods, such as fruits, vegetables, and fish.

Based on the MEDAS score, the studied population was divided into three subgroups as shown in Table 3. The results indicate that 73.5% of participants showed a moderate adherence to the Mediterranean diet, with only 11.7% of the population reaching a high MEDAS score (score ≥ 10). Similar results have also been observed by Di Renzo et al. [26], who reported, in a similar Italian survey, a high MEDAS score in only 15.3% of the assayed population.

The Food and Agriculture Organization (FAO) of the United Nations states that maintaining a healthy diet is an important part of supporting a strong immune system [32]. Therefore, a correct diet that is rich in nutrients, with strong antioxidant and anti-inflammatory properties, such as that suitable for the MD, could help to restore the immune system, reducing the virulence of severe acute respiratory syndrome coronavirus 2 (SARS-CoV-2) [26].

### 3.3. Lifestyle and Eating Habit Changes during COVID-19 Emergency

The rapid, global spread of COVID-19 has led to enormous challenges in the health system, the economy, and food supply, globally and locally. The advent of the COVID-19 lockdown, and subsequent measures, have substantially changed lifestyles in the Italian population. Survey respondents reported changes in their eating habits during the global pandemic period. Data reported a minimal influence (5–20%) for 33.5% of participants, whereas 26.1% of participants reported an impact of 20–50%. Results in the third part of the questionnaire regarding lifestyle and eating habit changes are reported in Appendix A.

During the pandemic period, participants reported cereal consumption at least once a day (60.1% of cases), and in 22.6% of cases, the daily cereal consumption was twice a day. The daily consumption of cheese and dairy products was twice a day in 32.3% of participants, while 44.7% replied that they consumed cheese and dairy products with a frequency of 2–3 times a week. As for egg consumption, 74.7% consumed on average of twice per week, whereas 9.3% declared not consuming eggs. In regards to the consumption of wholegrain products interesting data were obtained: 34% consumed daily wholegrain products, while 26.1% declared not to have eaten them during the pandemic period.

To ensure good nutritional status during a lockdown, it appears essential to check an individual’s habits in order to suggest a correction diet [33]. It is widely demonstrated that adherence to the Mediterranean diet is able to reduce the occurrence of numerous human diseases. A balanced diet rich in bioactive compounds, mostly present in fresh plant products, positively influences the body. As demonstrated by a cumulative analysis among eight cohorts (*n* = 514,816), which evaluated the relationship between the adherence to a Mediterranean diet and the mortality risk, a reduced risk of 0.91 (95% confidence interval 0.89 to 0.94) was observed. Likewise, a beneficial role for greater adherence to a Mediterranean diet on cardiovascular mortality was shown (pooled relative risk 0.91, 0.87 to 0.95), the incidence from cancer (0.94, 0.92 to 0.96), and incidence of Parkinson’s and Alzheimer’s disease (0.87, 0.80 to 0.96) [34].

The guidelines for a healthy and correct Italian diet are reported by the Ministry of Health [35]. In addition, with the IV Revision of the Reference Intake Levels of Nutrients and Energy for the Italian Population (LARN), the Italian Society of Human Nutrition (SINU) provides a nutritional document that can be used for nutritional planning in a single individual [36]. The IV revision of Nutrients and Energy for Italian population (LARN) is the document of the Italian Society of Human Nutrition (SINU).

Moreover, since isolation impact could lead to greater accessibility to food consumption, participants were asked how many meals they used to eat per day. The diversification of answers was in line with a good percentage of people who regularly consumed five meals a day (32.6%), whereas only 5.8% of the participants stated to have only two meals a day (lunch and dinner). However, 69.8% of participants regularly ate breakfast, while 3.4% declared they did not eat it. Among snacks, fruits were the most commonly consumed snack in the present survey (40.2%). Eight percent of the participants declared consuming junk food as snacks, and approximately 24.5% stated that they did not have snacks during the day. In regards to the “Food rude” (Junk food) increase, 35% of participants reported an increase of 5–20% and 22.5% of respondents reported an increase of 20–50% in. In addition, in 55.9% of responses, an increased consumption of sweet food was reported. 

“Food rude” refers to consuming food that is poor in essential nutrients, energy-dense, and less healthy compared to fresh meals [37]. Several studies reported that the high consumption of such foods contributes to weight gain and increased risks of chronic disease, including cardiovascular disorder and obesity [38,39]. Based on large cohort data reported by a survey conducted in China [40], it seems that adults with more-severe obesity (larger body mass index, BMI) and those with central obesity (larger waist circumference or higher waist-to-hip ratio) are at a higher risk for developing severe COVID-19 infections. However, weight gain has been recorded by several studies conducted during the lockdown period: an Italian survey, which involved a large number of participants (*n* = 3533), showed that 53.9% of participants have changed their lifestyles, and junk food and sweets increased in two-thirds of cases compared to the usual intake. Moreover, the perception of weight gain was recorded in 48.6% of the responders [11]. In a survey conducted in India [41], the reduced intake of fast food, fried food, junk food sweets, and chocolates was reported by around 20% of the participants (*n* = 103). In another study, conducted in India, patients (*n* = 6168) affected by type 2 diabetes mellitus, reported increased levels of junk food, carbohydrate intake, weight gain, and destabilized glucose control [42]. Based on the evidence, the World Health Organization (WHO) suggested dietary guidelines during the COVID-19 period, stressing the importance of an equilibrate diet to support the immune system and to minimize chronic diseases [43].

In regards to frozen product consumption, it emerged that there was an increase in 81.3% of responses of the survey. On the consumer side, there was an instinctive reaction to hoard food, with the trend to buy conservable products. During the initial phase of the pandemic period, there was a move towards fourth and fifth range products (readymade foods). The latest report by the ‘Istituto di Servizi per il Mercato Agricolo Alimentare’ (Ismea) [12] states that, in the weeks from 17 February to 15 March, Italian citizen spending on packaged products exceeded that of the previous four weeks by 17%, and that of the same weeks in 2019 by 19%. During the COVID lockdown, the average weekly food purchase of an Italian citizen increased compared with normal household consumption. The lockdown period generated changes in consumer habits and, consequentially, in the quality of their diets. An orientation towards the consumption of processed food was observed, including junk foods, snacks, and ready-to-eat cereals, convenience foods, and a reduction in health foods [44]. A cohort study carried out in France during the pandemic period also reported a reduction in buying fresh products, due to less possibility of having access to shops. This led to increases in frozen and canned vegetable intake, decreasing the overall possibility of taking bioactive compounds from fresh vegetables, which have a key role in preventing the onset of diseases, or protecting against diseases [45]. As reported by Di Renzo et al. [11], during the COVID-19 lockdown, more than 30% of the participants declared eating more or less healthy food.

Among the numerous aspects in which to pay attention, when it comes to correct lifestyles and healthy nutrition, the methods of cooking food deserve special attention. In fact, considering that, for a large variety of foods, cooking is a fundamental prerequisite of edibility, together with the choice of raw materials, the cooking method must be well chosen to ensure healthiness, safety, and taste. In this context, a positive aspect seems to be related to the cooking methods applied during the pandemic period. In fact, when participants were interviewed about the most used cooking methods, baking was the favored one (47.1%), while frying represented only 2.4% of the results. In particular, the frying method was used only once a week by 52.9% of the participants. Nonetheless, our study also revealed that the lockdown situation also created an opportunity to improve nutritional behaviors, such as cooking homemade meals, probably due to the inability to order take away food.

In terms of beverages, responses highlight a low water consumption, which was less than 1 L for 33.1% of participants, whereas 4% declared a water consumption of less than 500 mL. Natural water was the most consumed (88.8%). For 91% of participants there was no increase in the consumption of fruit juices. Coffee intake increased for 64.8% of participants, 17.7% of participants were not coffee consumers. Regarding alcohol consumption, 15% of subjects said they did not experience a change in alcohol intake during the pandemic period, only 4% declared an increase. Alcohol consumption during the pandemic period was twice per week for 26.1% of participants, 1.9% consumed it daily.

In regards to changes in body weight, 58.8% of the interviewees reported a variation. In particular, 40.6% underwent a weight gain of 1–3 kg, whereas a weight gain of 3–7 kg was recorded for 12.5% of the respondents. It should be noted that 9% of the survey participants experienced a decrease in their body weight of 1–2 kg. In addition, 70.5% of participants declared a decrease physical activity during the pandemic period: 14.3% perform physical activity for one hour a day, 13.6% for one hour a week, 23.9% for three hours a week, while 40.8% did not participate in any physical activity. Our results were in line with another Italian survey conducted by Di Renzo et al. [11], in which 37.4% of the study population declared a stable weight, 13.9% believed they lost weight, 40.3% felt they had a slight weight gain, and 8.3% gained a lot of weight. Substantial changes in outdoor time (93.6%) were reported also by Balanzá-Martínez et al., who completed a survey involving 1254 Spanish individuals. In particular, 70.2% (*n* = 879) reported change in physical activity, 34.7% (*n* = 433) in stress management, 33.7% (*n* = 414) in social support, 37.3% (*n* = 467) in sleep patterns, 23.4% (*n* = 294) in diet and nutrition [46].

Violin plots were used to represent comparison of the age distribution across different eating habits (such as attitude in having breakfast, the consumption of frozen foods, cooking types, water intake, alcohol consumption, and being physically active. Considering violin plots in Figure 2, it was observed that children and adolescents (0–17 years old), regularly consumed breakfast (Figure 2A), but they were sensitive to eating frozen foods (Figure 2B), preferring oven cooked (Figure 2C). In a non-comforting scenario, it is clear that the very young (0–17 years old) drank little water, and consumed not-insignificant quantities of alcohol during the pandemic (Figure 2D,E). Breakfast was daily consumed by most people in the groups of the different age ranges considered; whereas skinnier sections evidenced that people aged 18–30 stated that they sometimes or rarely had this important meal (Figure 2A). Steamed food was preferred by adults over 55, and young people experienced griddle cooking (Figure 2C). Although, physical activity improved overall well-being, it was poorly practiced by adult participants (Figure 2F).

Furthermore, using the descriptive statistics module in JASP, a scatter plot of the data was created, with the x-axis—eating habit percentage changes or educational level, and on the y-axis—the weight change (%), and with different colors for male and female participants (by adding gender as the split variable). As depicted in Figure 3A, when eating habits largely varied (from 20 to 50%), a moderate weight gain (plus 1–3 kg) mostly affected female people, while males were also fattened by 3–7 kg following a variation in their eating habits, ranging between 50 and 80%. Exploring the relationship between education and body weight changes (Figure 3B), it appears that people with a lower level of education were gaining or losing weight more than those with degree qualification.

Appendix A reports scatter plots of the other data, also considering gender as a split variable. Age is on the x-axis, whereas the y-axis includes meal number (panel A), snack type (panel B), coffee consumption increase (panel C), fruit juice consumption (panel D), cereal portion (panel E), whole grain consumption (panel F), sweets consumption increase (panel G), eggs consumption (panel H). The age group divided by gender was: 0–17 (4 male and 20 female); 18–30 (205 male and 617 female); 31–55 (203 male and 430 female) >55 (20 male and 20 female). It was observed that children and adolescents consumed sweet products differently, with male participants attesting to eating more sweets. On the other hand, female young people appeared to consume the highest variety of snacks.

Thus, a trend towards rather unfavorable nutritional behaviors was observed: decreased physical activity levels and increased sedentary time, and eventually weight gain. The weight gained may become permanent if the unfavorable nutritional behaviors are not reversed. This is of concern, especially for individuals who were already overweight or obese at the beginning of the lockdown. On the contrary, some participants considered the lockdown period as an opportunity to spend more time cooking and to balance their overall diets. Therefore, a detailed vigilance is required to assess individual behavior at the end of the COVID-19 pandemic.

## 4. Conclusions

The current study gave us several clues on how the COVID-19 pandemic period has affected nutritional behaviors. Data reported in the current study suggested that during the lockdown period a substantial part of the population had unhealthy nutritional behaviors, which could be increase, in the long term, the risk of several diseases.

The results indicate that 73.5% of participants showed a moderate adherence to the Mediterranean diet, which could represent one of the best food protocols as the adjuvant therapeutic choice of COVID-19. Changes in eating habits, such as an increase in the consumption of frozen products (81.3%), coffee intake increase (64.8%), and sweet foods increase (55.9%) were observed. The reduction in alcohol consumption was declared by 81% of participants. Positive weight change was declared for 58.8% (40.6% underwent an increase of +1–3 kg). In addition, 70.5% of participants declared a decrease in physical activity during the pandemic period.

The questionnaire was a valid and suitable tool to assess, monitor, and record the dietary and lifestyle changes declared by participants during the COVID pandemic, although with natural limitations—representing only one part of the population—the one with more access to technology. The limitations of our study are mainly related to the possible selection bias, as reported data collected using social networks typically suffer from self-reporting biases.

Moreover, further surveys could be helpful to monitor lifestyle changes around the world during this emergency period, in order to provide critical suggestions to public health policymakers, to suggest (eventual) dietary advice to the population, to better face the end of the pandemic period.

## Figures and Tables

**Figure 1 nutrients-13-01197-f001:**
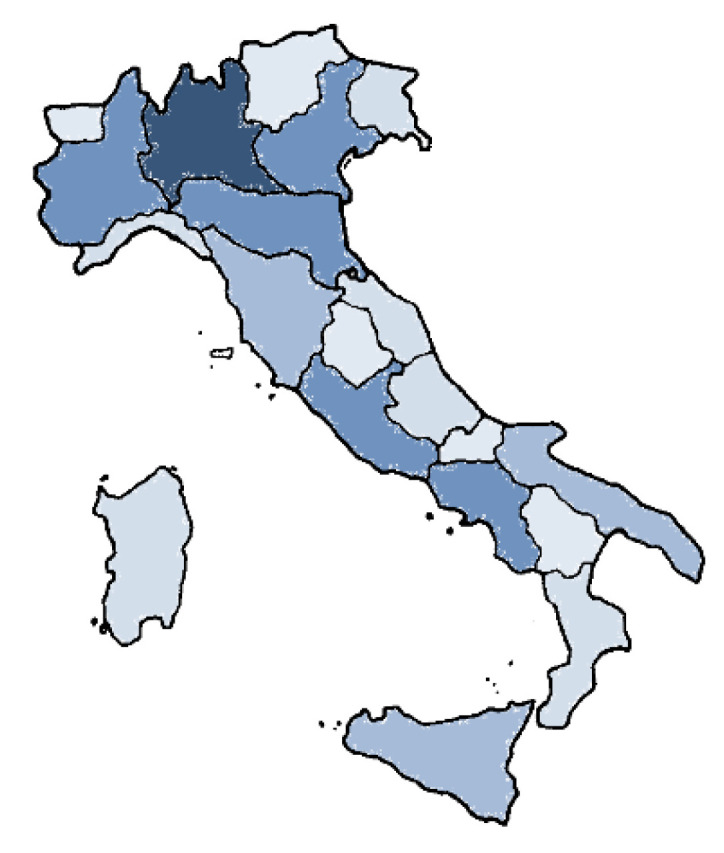
Cumulative incidence of coronavirus disease 2019 (COVID-19) infections by Italian Region (data upload 17 January, 2021). The different colors indicate the density of ascertained cases: dark blue corresponds to a maximum level of contagion, and light blue, a minimum level of contagion.

**Figure 2 nutrients-13-01197-f002:**
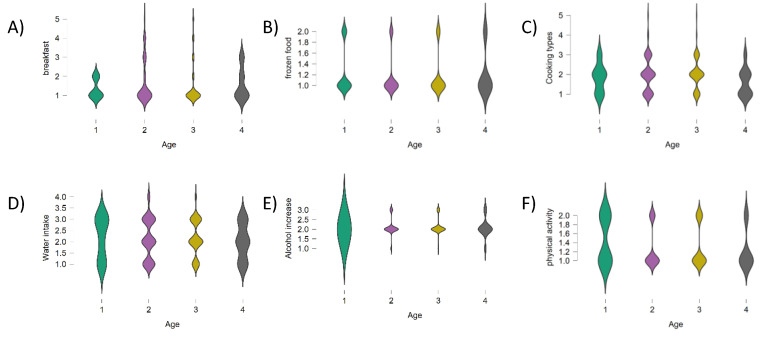
Violin plots created with JASP (www.jasp-stats.org, accessed on 20 January 2021), distinguishing between four different age groups (1 = 0–17; 2 = 18–30; 3 = 31–55; 4 = >55) within (**A**) breakfast consumption (1 = daily; 2 = often; 3 = sometimes; 4 = rarely; 5 = never); (**B**) frozen food intake (1 = yes; 2 = no); (**C**) cooking types (1 = steam; 2 = oven; 3 = griddle; 4 = frying; 5 = microwave); (**D**) water intake (1 = yes; 2 = no); (**E**) alcohol increase consumption in respect to pre-Covid-19 period (1 = yes; 2 = no; 3 = no variation); (**F**) physical activity practice (1 = yes; 2 = no).

**Figure 3 nutrients-13-01197-f003:**
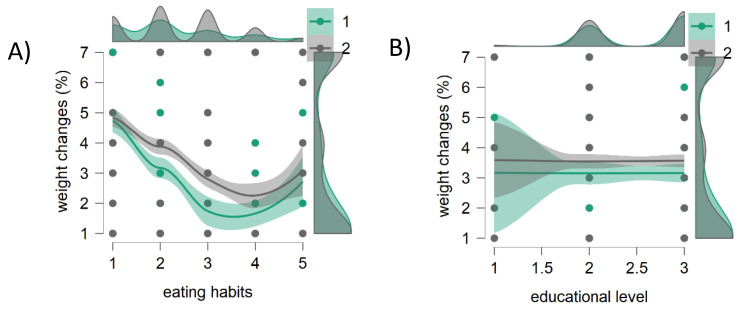
Scatter plots created with JASP (www.jasp-stats.org, accessed on 20 January 2021). (**A**) The x-axis eating habits, y-axis weight changes (%); (**B**) x-axis educational level, y-axis weight changes (%). The lines for the relation between eating habits and weight changes (%) are different for female participants (1●), and male participants (2●). The relation between educational level and weight changes (%) is flat for both the genders. Eating habits is for the percentage variation of individual eating habits induced by the lockdown (1 = 0–5%; 2 = 5–20%; 3 = 20–50%; 4) 50–80%; 5 = > 80%); three educational levels were considered (1 = secondary school certificate; 2 = high school diploma; 3 = degree-master-PhD); weight changes were as follows: 1 = ↑1–3 kg; 2 = ↑3–7 kg; 3 = ↑>8 kg; 4 = ↓1–3 kg; 5 = ↓3–5 kg; 6 = ↓>6 kg; 7 = no variation).

**Table 1 nutrients-13-01197-t001:** General characteristics of the study population and Mediterranean Diet Adherence Screener (MEDAS) score by gender.

	Participants (*n* = 1519)		MEDAS Score (Mean) ± SD	
	Male	Female	Male	Female
Participants	432 (28.4)	1087 (71.6)	7.4 ± 1.8	7.4 ± 1.8
Age groups (year)				
0–17	4 (0.3)	20 (1.3)	7.0 ± 2.0	7.6 ± 2.4
18–30	205 (13.5)	617 (40.6)	7.9 ± 1.7	7.8 ± 1.8
31–55	203 (13.4)	430 (20.3)	6.8 ± 1.7	6.9 ± 1.8
>55	20 (1.3)	20 (1.3)	7.1 ± 1.9	6.7 ± 1.6
Level of education				
Secondary school certificate	6 (0.4)	17 (1.1)	4.7 ± 2.7	4.2 ± 1.7
High school	173 (11.4)	457 (30.1)	6.2 ± 1.4	6.3 ± 1.4
Degree—masters or PhD	253 (16.7)	613 (40.3)	8.3 ± 1.5	8.4 ± 1.5

Values are expressed as number and percentage in parenthesis (*n* (%)) and as mean ± standard deviation (SD).

**Table 2 nutrients-13-01197-t002:** Positive answer to MEDAS questionnaire.

MEDAS Item	Whole Sample
	*n* = 1519
Olive oil, culinary fat	1411 (92.9)
Olive oil, ≥4 ts */day	712 (46.9)
Vegetables, ≥1 s */day	796 (52.4)
Fruits, ≥3 s/day	145 (9.6)
Read meat, <1 s/day	1281 (84.3)
Butter, <1 s/day	698 (46)
Sweet beverage, <1 s/day	1280 (84.3)
Wine, 7 s/week	129 (8.5)
Legumes, ≥3 s/week	985 (64.8)
Fish and seafood, ≥3 s/week	537 (35.4)
Sweets, <3 s/week	726 (47.8)
Nuts, ≥3 s/week	797 (52.5)
White meat over red	984 (64.8)
“Soffritto”	821 (54.0)

* ts: tablespoons; s: serving. Data are expressed as number and percentage in parenthesis (*n* (%)).

**Table 3 nutrients-13-01197-t003:** Adherence to the Mediterranean Diet (MD).

Adherence to the MD	*n* (%)
Low	224 (14.7)
Moderate	1117 (73.5)
High	178 (11.7)

The studied population was divided into three classes: low (score ≤ 5), moderate (score between 6 and 9), and high (score ≥ 10) adherence to the Mediterranean diet. Values are expressed as number and percentage in parenthesis (*n* (%)).

## Data Availability

The data presented in this study are available on request from the corresponding author.

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
