# Peer review of "An Italian Survey on Dietary Habits and Changes during the COVID-19 Lockdown"

_nutrients, 2021, doi:10.3390/nu13041197_

Round 1

Reviewer 1 Report

Overall the article presents novel and important findings.  There are some areas where it can be improved.  The major issue is the handling of the gender imbalance with over 70% of participants being female.  A useful analysis would be to analyze the genders separately and see if findings apply to both genders.  Also in the section of differences in age groups and education a gender factor should be included.  

Specific changes beyond this are outlined below

Line 26-27

Suggest changing the following to highlight the important and counterintuitive finding of alcohol decrease from:

" Interestingly, 81.3% of responders reported an increase in frozen food consumption, and alcohol assumption decreased for 81% of responders."

To:  A decrease in alcohol consumption was reported by 81% of responders, while an increase in frozen food consumption was reported by 81.3% of responders.

Line 31: change

 "although pandemic is yet ongoing" to "although the pandemic is yet ongoing"

Line 42-45 

Delete this sentence.  The origin of the disease is highly controversial, most likely zoonotic and not relevant to this paper.  The WHO report https://www.who.int/publications/m/item/who-convened-global-study-of-the-origins-of-sars-cov-2

Line 54-55  Update population numbers with more recent numbers (given is 1/17/21)

Table 1 should also be updated with most recent numbers as these have increased substantially

Line 114-121 

Data on the severity and incidence of serious COVID-19 in people with obesity should be added to this section.  See: Covid-19: Highest death rates seen in countries with most overweight populations BMJ 2021; 372 doi: https://doi-org.ezproxy.cul.columbia.edu/10.1136/bmj.n623 (Published 04 March 2021) Cite this as: BMJ 2021;372:n623

Line 164 was should be were

Line 162-170  Mention needs to be made of the extremely unbalanced gender differences in the population (72% female; 28% male) and how this might effect outcomes

Line 338 change "positive" to "increase" as positive could be interpreted as "beneficial" which is the opposite meaning of what is intended

Line 340 The reduction in alcohol consumption is clearly attributable to the im-340 possibility of peoples to go out and socialize--   This is only speculation – no data is provided to substantiate this statement – thus it should be deleted

Line 342-343 "An expected data was about sweet food, whose intake increased in 55.9% of cases."    Need clarification of what this sentence is trying to say.

Lines 400-415 need to show the N for male and female of each age group given the extreme imbalance in gender in the study population

Author Response

Response to Reviewer 1 Comments

Manuscript ID: nutrients-1149316

Title: An Italian survey on dietary habits changes during the COVID-19 lockdown

Comments and Suggestions for Authors

Overall the article presents novel and important findings.  There are some areas where it can be improved.  The major issue is the handling of the gender imbalance with over 70% of participants being female.  A useful analysis would be to analyze the genders separately and see if findings apply to both genders.  Also in the section of differences in age groups and education a gender factor should be included.  

As suggested by reviewer 1, the authors improved this aspect. In Table 2 the background characteristics of participants by gender and differences in age groups and education were presented.

 Specific changes beyond this are outlined below

 Point 1: Line 26-27

Suggest changing the following to highlight the important and counterintuitive finding of alcohol decrease from:

" Interestingly, 81.3% of responders reported an increase in frozen food consumption, and alcohol assumption decreased for 81% of responders."

To:  A decrease in alcohol consumption was reported by 81% of responders, while an increase in frozen food consumption was reported by 81.3% of responders.

Response 1:  As suggested by reviewer 1, the authors change the sentence " Interestingly, 81.3% of responders reported an increase in frozen food consumption, and alcohol assumption decreased for 81% of responders." as “A decrease in alcohol consumption was reported by 81% of responders, while an increase in frozen food consumption was reported by 81.3% of responders.”

Point 2: Line 31: change

 "although pandemic is yet ongoing" to "although the pandemic is yet ongoing"

Response 2:  As suggested by reviewer 1, the authors changed "although pandemic is yet ongoing" as "although the pandemic is yet ongoing"

Point 3: Line 42-45 

Delete this sentence.  The origin of the disease is highly controversial, most likely zoonotic and not relevant to this paper.  The WHO report https://www.who.int/publications/m/item/who-convened-global-study-of-the-origins-of-sars-cov-2

 Response 3:  As suggested by reviewer 1, the authors deleted this sentence.

Point 4:  Line 54-55 Update population numbers with more recent numbers (given is 1/17/21)

Table 1 should also be updated with most recent numbers as these have increased substantially

Response 4:  As suggested also by reviewer 3, the authors deleted Table 1 because the COVID cases and deaths in different parts of the world rapidly changing and it does not correlated with the objective of the paper.

Point 5: Line 114-121 

Data on the severity and incidence of serious COVID-19 in people with obesity should be added to this section.  See: Covid-19: Highest death rates seen in countries with most overweight populations BMJ 2021; 372 doi: https://doi-org.ezproxy.cul.columbia.edu/10.1136/bmj.n623 (Published 04 March 2021) Cite this as: BMJ 2021;372:n623

Response 5:  As suggested by reviewer 3, the authors added the correlation among covid-19 mortality and the proportion of adults that are overweight.

Point 6: Line 164 was should be were

Response 6:  As suggested by reviewer 1, the authors changed “was” as “were”.

Point 7: Line 162-170  Mention needs to be made of the extremely unbalanced gender differences in the population (72% female; 28% male) and how this might effect outcomes

Response 7: As suggested by reviewer 1, the authors added the missing information as “Of the total participants 71.6% were females, to overcome the unbalanced gender differences in the studied population, the background characteristics of participants were divided by gender.”

Point 8: Line 338 change "positive" to "increase" as positive could be interpreted as "beneficial" which is the opposite meaning of what is intended

Response 8: As suggested by reviewer 1, the authors changed “positive” as “increase”.

Point 9: Line 340 The reduction in alcohol consumption is clearly attributable to the im-340 possibility of peoples to go out and socialize--   This is only speculation – no data is provided to substantiate this statement – thus it should be deleted

Response 9: As suggested by reviewer 1, the authors deleted this sentence.

Point 10: Line 342-343 "An expected data was about sweet food, whose intake increased in 55.9% of cases."    Need clarification of what this sentence is trying to say.

Response 10: As suggested by reviewer 1, the authors clarify this sentence.

Point 11: Lines 400-415 need to show the N for male and female of each age group given the extreme imbalance in gender in the study population

Response 11: As suggested by reviewer 1, the authors added the number of male and female of each age group as ” The age group divided by gender was: 0-17 (4 male and 20 female:); 18-30 (205 male and 617 female); 31-55 (203 male and 430 female) >55 (20 male and 20 female).”

The authors thank the reviewer for the valuable suggestions that allowed us to improve the manuscript.

Reviewer 2 Report

The manuscript “An Italian survey on dietary habits changes during the COVID-19 lockdown” presents results from a study performed during the 1st phase of pandemic, in Italy, the European country most affected in the first times of pandemics. This presents relevant information about food habits in that period and has the novelty of searching for Mediterranean Diet adherence level, in particular.

The manuscript has some points that need to be addressed.

Abstract

Line 28 – Please remove “although”. I think this word makes no sense here, since the weight gain is in line with decrease physical activity.

Introduction

Line 41 – Please delete “unknown causes”. At this moment, this is not completely true.

Line 49 – “can occur also by contact with contaminated objects or surfaces” – Please be careful, since this is not consensual. I suggest to delete since besides the lack of strong evidences about the contamination by surface contact, this is not necessary in the context of the study.

Line 121 – Since Mediterranean Diet was focus of the study, more information about this dietary pattern/lifestyle should be included. In fact, much of the text from the beginning of discussion makes more sense here, in the introduction.

Material and methods

Line 139 – I suggest to delete this first sentence

Results and discussion

Line 164 – “were divided into four different age groups” – please specify the groups

Table 2 – In the column of MEDAS score, please add if this is standard deviation or standard error

Point 3.2, until line 207 – I think much of this text can be passed to introduction.

Lines 207-209 – The authors are referring to this study or to studies from other authors. If is the second case, please add references. This needs to be clarified.

Line 285 – please delete “however”

Lines 288-289 – The sentences that come after are not about weight changes, but about intake changes. So, this needs to be re-written to make more sense to support the results about weigh changes on references reporting intake changes.

Lines 340-342 – this needs to be supported by a reference to other studies.

Line 387 – The percentages of changes in eating habits considered for analysis represent changes in a positive way (toward healthier habits) and in a negative way, simultaneously? Moreover, it is not clear the parameters considered to calculate this percentage. All this needs to be clarified in material and methods.

Conclusions – I think authors also need to add limitations. For example the fact of having the study based on a online survey, with the natural limitations of representing only one part of population (the one with more access to technologies). Moreover, from what I could understand, the questionnaire was not a previously validated questionnaire. In my point-of view, this does not affect the relevance of this study, but needs to be taking into consideration.

Author Response

Response to Reviewer 2 Comments

Manuscript ID: nutrients-1149316

Title: An Italian survey on dietary habits changes during the COVID-19 lockdown

Comments and Suggestions for Authors

The manuscript “An Italian survey on dietary habits changes during the COVID-19 lockdown” presents results from a study performed during the 1st phase of pandemic, in Italy, the European country most affected in the first times of pandemics. This presents relevant information about food habits in that period and has the novelty of searching for Mediterranean Diet adherence level, in particular.

 The manuscript has some points that need to be addressed.

 Abstract

Point 1: Line 28 – Please remove “although”. I think this word makes no sense here, since the weight gain is in line with decrease physical activity.

Response 1: As rightly suggested by reviewer 2, the authors removed the improper word.

Point 2: Introduction

Line 41 – Please delete “unknown causes”. At this moment, this is not completely true.

Response 2: As rightly suggested by reviewer 2, the authors deleted this sentence.

Point 3: Line 49 – “can occur also by contact with contaminated objects or surfaces” – Please be careful, since this is not consensual. I suggest to delete since besides the lack of strong evidences about the contamination by surface contact, this is not necessary in the context of the study.

Response 3: As suggested by reviewer 2, the authors deleted this sentence.

Point 4: Line 121 – Since Mediterranean Diet was focus of the study, more information about this dietary pattern/lifestyle should be included. In fact, much of the text from the beginning of discussion makes more sense here, in the introduction.

Response 4: As suggested by reviewer 2, the authors added more information about the dietary MD model. Parts of text included in the first part of the discussion section were moved to the introduction section.

Point 5: Material and methods

Line 139 – I suggest to delete this first sentence

Response 5: As suggested by reviewer 2, the authors deleted this sentence.

Point 6:

Results and discussion

Line 164 – “were divided into four different age groups” – please specify the groups

Response 6: As suggested by reviewer 2, the authors added the missing information.

Point 7: Table 2 – In the column of MEDAS score, please add if this is standard deviation or standard error

Response 7: The reported value referred to the standard deviation. As suggested by reviewer 2, the authors added the missing information.

Point 8: Point 3.2, until line 207 – I think much of this text can be passed to introduction.

Response 8: As suggested by reviewer 2, the authors moved this part to the introduction section.

Point 9: Lines 207-209 – The authors are referring to this study or to studies from other authors. If is the second case, please add references. This needs to be clarified.

Response 9: The authors are referring to this study and clarify this sentence in the manuscript.

Point 10: Line 285 – please delete “however”

Response 10: As suggested by reviewer 2, the authors deleted the word “however” in line 285.

Point 11: Lines 288-289 – The sentences that come after are not about weight changes, but about intake changes. So, this needs to be re-written to make more sense to support the results about weigh changes on references reporting intake changes.

Response 11: As suggested by reviewer 2, the authors modified this sentence and added the missing information.

Point 12: Lines 340-342 – this needs to be supported by a reference to other studies.

Response 12: As also suggested by reviewer 1, the authors deleted this sentence.

Point 13: Line 387 – The percentages of changes in eating habits considered for analysis represent changes in a positive way (toward healthier habits) and in a negative way, simultaneously? Moreover, it is not clear the parameters considered to calculate this percentage. All this needs to be clarified in material and methods.

Response 13: Responses to answer number 38 “Did your eating habits changed during the COVID19 pandemic period?” in our Questionnaire (please see Appendix) are used for statistical purposes. This is indicated at line 381, where it is written: “with on the x-axis eating habits percentage changes”. Participants who declared to vary their eating habits by 0-5% (1 in x-axis), favourably lose 1-3 kg; whereas an eating habits change equal to 20-50% (3 in x-axis) and 50-80% (4 in x-axis) was consistent in gain weight. When female participants vary their eating habits by 20-25%, a gain 3-7 kg was observed.

Point 14: Conclusions – I think authors also need to add limitations. For example the fact of having the study based on a online survey, with the natural limitations of representing only one part of population (the one with more access to technologies). Moreover, from what I could understand, the questionnaire was not a previously validated questionnaire. In my point-of view, this does not affect the relevance of this study, but needs to be taking into consideration.

Response 14: As suggested by reviewer 2, the authors added the limitations of the online questionnaire in the conclusions section.

The authors thank the reviewer for the valuable suggestions that allowed us to improve the manuscript.

Reviewer 3 Report

General comments: all the manuscript need a throughout review of the English language.

My main concern with this paper is that it measures MD score at present and then the survey asks for changes in some food items during the lockdown, but the measures are not comparable. So it is difficult to assess the changes and to make any type of comparison.

Abstract: in the sentence “In addition, 58.8% reported positive weight modification (40.8%, +1-3kg), although 28 physical activity reduction was reported for 70.5% of responders”, I would change the “although”, since the weight increase might be related to PA reduction, so it is not a different statement.

Introduction: as a general comment for the introduction, I would encourage the authors to reduce it since a lot of the information presented is not relevant at all for the paper. Specific comments below.

 “Since December 2019, the entire world is facing a global challenge against the SARS-Cov-2 which is associated with a severe acute respiratory syndrome of unknown cause (COVID-19),”. I would not state that COVID-10 is of unknown cause

“To 49 avoid catch COVID-19 infection some behaviours were recommended to residents including keeping social distancing, maintain at least 1 meter away from others, wearing safety devices (masks and gloves) and frequent cleaning hands and their disinfection with specific alcoholic gels were strongly recommended [6].” I don’t think this sentence is needed, since it does not have anything to do with the objective of the paper. Also no need to report the COVID cases and deaths in different parts of the world, it is an unnecessary table. This figures are rapidly changing and do not add anything to this paper in particular.

Figure 1: can you add a more specific legend?

Line 89 “smart working”: one of the definitions of smart working is ”a new model of work that uses the new technologies and the development of existing technologies to improve both the performance and the satisfaction that is obtained from the job”. I think that when the pandemic started, most people moved to teleworking, which is an essential part of smart working, but the latter term implies much more things, and I don’t think that most companies were ready for this at that point in time.

Methods: how is the population who participated in the survey comparable to the Italian population? As stated, the survey was distributed using social networks, so this will have a clear selection bias.

Line 141: could you please explain what “subjective statements” are?

Results and discussion:

Line 164-165: “Participants were divided into four different age groups: 54.1% was in the age group 18-30 and 41.7% in the age group 31-54”. According to the text, 4 age groups were created, but only 2 are reported which is misleading.

How was the consent form obtained? Especially for the 0-17 year age group. Which was the minimum age of the participants? Again, reporting the age group of 0-17 is misleading. And for the older age group, which was the maximum age of participants?

Lines 166-168: are the MEDAS scores by age different statistically significant? The p-value showed is analysing differences between the 4 groups? If not the statement is not correct. And also, taking into account the sample size.

Tables 2 and 3 can be merged into one table. Table 3 “diploma” has to be put within the same line as “high school”.

Paragraph from line 180 to 191: it is very confusing. Not clear what comorbidities were asked in the survey, what comes from published statistics… And the only comparison made is with some comorbidities and education, but nothing related to MEDAS. Why are these comorbidities asked and not others? The list seems quite short.

Section 3.2: it includes some information that need to be included in the introduction, since it is not results neither discussion of the present paper. The same for section 3.3.

Table 4 should be before Tables 2 and 3, since it describes the MEDAS score. Also, first item on MEDAS score ask for olive oil for as main culinary fat, but on Table 4 is reported as “dressing”, which is a different concept.

Table 5. The second column, according to the title, shows the MEDAS score, which is not possible. I guess this is the number of subjects?

Lines 245-247: how is this statement assessed?

Section 3.3: the information reported is quite messy. There is no clear link/comparison between the results reported in the present study and the references in the literature. There are some statements about consumer behaviours without any citation and it is not some information that has been collected with the survey.

Author Response

Response to Reviewer 3 Comments

Manuscript ID: nutrients-1149316

Title: An Italian survey on dietary habits changes during the COVID-19 lockdown

Point 1: General comments: all the manuscript need a throughout review of the English language. My main concern with this paper is that it measures MD score at present and then the survey asks for changes in some food items during the lockdown, but the measures are not comparable. So it is difficult to assess the changes and to make any type of comparison.

Response 1: As rightly suggested by reviewer 3, the authors checked the English language for the entire manuscript. In this work, the MEDAS score was used to investigate the adherence to the Mediterranean diet among the Italian population during the lockdown, whereas changes in eating habits during the lockdown were assessed.

Point 2: Abstract: in the sentence “In addition, 58.8% reported positive weight modification (40.8%, +1-3kg), although 28 physical activity reduction was reported for 70.5% of responders”, I would change the “although”, since the weight increase might be related to PA reduction, so it is not a different statement.

Response 2: As rightly suggested by reviewer 3, the authors deleted “although” in the sentence in order to avoid misunderstandings.

Point 3: Introduction: as a general comment for the introduction, I would encourage the authors to reduce it since a lot of the information presented is not relevant at all for the paper. Specific comments below.

Response 3: As suggested by reviewer 3, the authors deleted not relevant information in the introduction section.

Point 4: “Since December 2019, the entire world is facing a global challenge against the SARS-Cov-2 which is associated with a severe acute respiratory syndrome of unknown cause (COVID-19),”. I would not state that COVID-10 is of unknown cause

Response 4: As suggested by reviewer 3, the authors deleted the state “unknown cause” from the sentence.

Point 5: “To 49 avoid catch COVID-19 infection some behaviours were recommended to residents including keeping social distancing, maintain at least 1 meter away from others, wearing safety devices (masks and gloves) and frequent cleaning hands and their disinfection with specific alcoholic gels were strongly recommended [6].” I don’t think this sentence is needed, since it does not have anything to do with the objective of the paper. Also no need to report the COVID cases and deaths in different parts of the world, it is an unnecessary table. This figures are rapidly changing and do not add anything to this paper in particular.2

Response 5: As suggested by reviewer 3, the authors eliminated unnecessary information. The authors also deleted table 1 reporting the COVID cases and deaths in different parts of the world.

Point 6: Figure 1: can you add a more specific legend?

Response 6: As suggested by reviewer 3, the authors modify the legend to Figure 1.

Point 7: Line 89 “smart working”: one of the definitions of smart working is ”a new model of work that uses the new technologies and the development of existing technologies to improve both the performance and the satisfaction that is obtained from the job”. I think that when the pandemic started, most people moved to teleworking, which is an essential part of smart working, but the latter term implies much more things, and I don’t think that most companies were ready for this at that point in time.

Response 7: As suggested by reviewer 3, the authors changed the word “smart working” as “teleworking”.

Point 8: Methods: how is the population who participated in the survey comparable to the Italian population? As stated, the survey was distributed using social networks, so this will have a clear selection bias.

Response 8: As suggested by reviewer 2, the authors added the missing information in the manuscript as "The limitations of our study are mainly related to the possible selection bias, as reported data collected using social networks typically suffer from self-reporting biases.”

Point 9: Line 141: could you please explain what “subjective statements” are?

Response 9: As suggested by reviewer 3, the authors clarify this sentence.

Point 10: Results and discussion: Line 164-165: “Participants were divided into four different age groups: 54.1% was in the age group 18-30 and 41.7% in the age group 31-54”. According to the text, 4 age groups were created, but only 2 are reported which is misleading.

Response 10: As also reported by another reviewer, this sentence has been clarified by the authors.

Point 11: How was the consent form obtained? Especially for the 0-17 year age group. Which was the minimum age of the participants? Again, reporting the age group of 0-17 is misleading. And for the older age group, which was the maximum age of participants?

Response 11: As rightly suggested by reviewer 3, the authors clarify this mislead part in the manuscript. The minimum age range of participants who were included in the range 0-17 was 12 years. For the older age group, the maximum age of the participants was 70 years.

Point 12: Lines 166-168: are the MEDAS scores by age different statistically significant? The p-value showed is analysing differences between the 4 groups? If not the statement is not correct. And also, taking into account the sample size.

Response 12: The authors confirm that the p-value referred to the differences between the 4 groups assayed, and the MEDAS score displayed by the group aged 18–30 years was statistically higher than the other groups.

Point 13: Tables 2 and 3 can be merged into one table. Table 3 “diploma” has to be put within the same line as “high school”.

Response 13: As suggested by reviewer 2, the authors merged Table 2 and 3 into one table and deleted the term diploma in the line Degree-Master-PhD, incorrectly indicated.

Point 14: Paragraph from line 180 to 191: it is very confusing. Not clear what comorbidities were asked in the survey, what comes from published statistics… And the only comparison made is with some comorbidities and education, but nothing related to MEDAS. Why are these comorbidities asked and not others? The list seems quite short.

Response 14: In the first part of the questionnaire, the authors asked the participants what their diagnosed pathologies were. The choice of comorbidities included six different comorbidities such as gastrointestinal, cardiovascular, thyroid, diabetes, cancer, and other diseases as a possibility choice, the main classes of pathologies reported by Italian statistics. The other diseases were represented only by 4.5%, highlighted that more than 95% of comorbidities were covered by the other five chosen comorbidities.

Point 15: Section 3.2: it includes some information that need to be included in the introduction, since it is not results neither discussion of the present paper. The same for section 3.3.

Response 15: As suggested by reviewer 3, the authors moved these parts in the introduction section.

Point 16: Table 4 should be before Tables 2 and 3, since it describes the MEDAS score. Also, first item on MEDAS score ask for olive oil for as main culinary fat, but on Table 4 is reported as “dressing”, which is a different concept.

Response 16: The authors preferred to leave Table 4 before Table 2 because the latter contains data on study design reported in the paragraph 3.1. As suggested by reviewer 3, the authors modify “Olive oil, main dressing” as “Olive oil, culinary fat”.

Point 17: Table 5. The second column, according to the title, shows the MEDAS score, which is not possible. I guess this is the number of subjects?

Response 17: As rightly noted by reviewer 3 there was an error in the second column, the authors changed the label "MEDAS Score" to "n (%)"

Point 18: Lines 245-247: how is this statement assessed?

Response 18: The statement assessed in these lines is referred to the result of question number 38 of the questionnaire.

Point 19: Section 3.3: the information reported is quite messy. There is no clear link/comparison between the results reported in the present study and the references in the literature. There are some statements about consumer behaviours without any citation and it is not some information that has been collected with the survey.

Response 19: As suggested by reviewer 3, the authors clarify this section.

The authors thank the reviewer for the valuable suggestions that allowed us to improve the manuscript.

Reviewer 4 Report

The aim of this study was to explore the impact of the COVID-19 pandemic on dietary habits, lifestyle changes, and adherence to the Mediterranean diet among the Italian population through an online questionnaire during the period of isolation due to COVID-19 and to explore its associated factors.

In view of COVID-2 19 pandemic situation the topic of work is very important and necessary. The study touches the important issue – the healthy diet may support the immune system and minimize chronic diseases during the COVID-19 period.

However, one concern is the title of present paper. The obtained results showed mainly the dietary habits in the studied group during the lockdown and additionally there were some questions connected with changes in these habits. Therefore, I think it should be divided on two sorts of information: An Italian survey on dietary habits and dietary habits changes during the COVID- 2 19 lockdown.

The authors used nonparapetric tests (Mann–Whitney U and Kruskal–Wallis tests) to compare continuous variables. According to the fact that normal distribution was found for all the variables it was possible to use parametric tests in these analyses.

Results about the third part of the questionnaire regard lifestyle and eating habits changes were reported in Figure S1 – is this Figure included?

Some stylistic errors were found, for example:

Line 277: Among the most consumed snacks, fruit represents the 40.2% of consumed snacks.

Were there any inclusion/exclusion criteria in this study? They should be presented.

Some limitations in this study should be described.

Author Response

Response to Reviewer 4 Comments

Manuscript ID: nutrients-1149316

Title: An Italian survey on dietary habits changes during the COVID-19 lockdown

The aim of this study was to explore the impact of the COVID-19 pandemic on dietary habits, lifestyle changes, and adherence to the Mediterranean diet among the Italian population through an online questionnaire during the period of isolation due to COVID-19 and to explore its associated factors. In view of COVID-2 19 pandemic situation the topic of work is very important and necessary. The study touches the important issue – the healthy diet may support the immune system and minimize chronic diseases during the COVID-19 period.

Point 1: However, one concern is the title of present paper. The obtained results showed mainly the dietary habits in the studied group during the lockdown and additionally there were some questions connected with changes in these habits. Therefore, I think it should be divided on two sorts of information: An Italian survey on dietary habits and dietary habits changes during the COVID- 2 19 lockdown.

Response 1: As suggested by reviewer 4, the authors changed the title of the article.

Point 2: The authors used nonparapetric tests (Mann–Whitney U and Kruskal–Wallis tests) to compare continuous variables. According to the fact that normal distribution was found for all the variables it was possible to use parametric tests in these analyses.

Response 2: As suggested by reviewer 4, since the normal distribution was found for all the variables, it was possible to use parametric tests in the analyses, however the authors used in data analysis nonparametric tests including Mann–Whitney U and Kruskal–Wallis tests.

Point 3: Results about the third part of the questionnaire regard lifestyle and eating habits changes were reported in Figure S1 – is this Figure included?

Response 3: Figure S1 was included in the supplementary material section.

Point 4: Some stylistic errors were found, for example: Line 277: Among the most consumed snacks, fruit represents the 40.2% of consumed snacks.

Response 4: As suggested by reviewer 4, the authors changed the sentence as “Among snacks, fruits were the most commonly consumed snack in the present studied survey (40.2%).”

Point 5: Were there any inclusion/exclusion criteria in this study? They should be presented. Some limitations in this study should be described.

Response 5: As suggested by reviewer 4, the authors added limitations of the questionnaire that resulted to be a valid and suitable tool to assess monitor and record the dietary and lifestyle changes declared by participants during the COVID pandemic although the natural limitations of representing only one part of the population, the one with more access to technologies.

The authors thank the reviewer for the valuable suggestions that allowed us to improve the manuscript.

Round 2

Reviewer 2 Report

The authors were able to address my previous concerns. I think the quality of the manuscript was improved and, at this point, the manuscript is acceptable in its present form.